# Epidemiology and burden of multidrug-resistant bacterial infection in a developing country

Cherry Lim[1†], Emi Takahashi[1†], Maliwan Hongsuwan[1], Vanaporn Wuthiekanun[1], Visanu Thamlikitkul[2], Soawapak Hinjoy[3], Nicholas PJ Day[1,4], Sharon J Peacock[1,5,6], Direk Limmathurotsakul[1,4,7*]

[1]Mahidol Oxford Tropical Medicine Research Unit, Faculty of Tropical Medicine, Mahidol University, Bangkok, Thailand; [2]Faculty of Medicine Siriraj Hospital, Mahidol University, Bangkok, Thailand; [3]Bureau of Epidemiology, Department of Disease Control, Ministry of Public Health, Nonthaburi, Thailand; [4]Centre for Tropical Medicine and Global Health, Nuffield Department of Medicine, University of Oxford, Oxford, United Kingdom; [5]London School of Hygiene and Tropical Medicine, London, United Kingdom; [6]University of Cambridge, Addenbrooke's Hospital, Cambridge, United Kingdom; [7]Department of Tropical Hygiene, Faculty of Tropical Medicine, Mahidol University, Bangkok, Thailand

*For correspondence: direk@ tropmedres.ac

[†]These authors contributed equally to this work

Competing interests: The authors declare that no competing interests exist.

**Abstract** Little is known about the excess mortality caused by multidrug-resistant (MDR) bacterial infection in low- and middle-income countries (LMICs). We retrospectively obtained microbiology laboratory and hospital databases of nine public hospitals in northeast Thailand from 2004 to 2010, and linked these with the national death registry to obtain the 30-day mortality outcome. The 30-day mortality in those with MDR community-acquired bacteraemia, healthcare-associated bacteraemia, and hospital-acquired bacteraemia were 35% (549/1555), 49% (247/500), and 53% (640/1198), respectively. We estimate that 19,122 of 45,209 (43%) deaths in patients with hospital-acquired infection due to MDR bacteria in Thailand in 2010 represented excess mortality caused by MDR. We demonstrate that national statistics on the epidemiology and burden of MDR in LMICs could be improved by integrating information from readily available databases. The prevalence and mortality attributable to MDR in Thailand are high. This is likely to reflect the situation in other LMICs.

## Introduction

The emergence of antimicrobial resistance (AMR) is of major medical concern, particularly in low- and middle-income countries (LMICs) (*World Health Organization, 2014*; *Laxminarayan et al., 2013*). In LMICs, antibiotic use is increasing with rising incomes, affordable antimicrobials and the lack of stewardship in hospital and poor control of over-the-counter sales. This is driving the emergence and spread of multidrug-resistant (MDR) pathogens in community and hospital settings. Hospital data from LMICs suggest that the cumulative incidence of community-acquired Extended-Spectrum Beta-Lactamase (ESBL) producing *Escherichia coli* and *Klebsiella pneumoniae* infections are increasing over time (*Kanoksil et al., 2013*; *Ansari et al., 2015*). A recent report from the International Nosocomial Infection Control Consortium (INICC) also showed that the prevalence of AMR organisms causing hospital-acquired infections (HAI) in ICUs in LMICs is much higher than those in the United States (US) (*Rosenthal et al., 2014*).

**eLife digest** Antimicrobial resistance is a global problem. Each year, an estimated 23,000 deaths in the United States and 25,000 deaths in the European Union are extra deaths caused by bacteria resistant to antibiotics. People in low- and middle-income countries are also using more antibiotics, in part because of rising incomes, lower costs of antibiotics, and a lack of control of antimicrobial usage in the hospitals and over-the-counter sales of the drugs. These factors are thought to be driving the development and spread of bacteria that are resistant to multiple antibiotics in countries such as China, India, Indonesia and Thailand. However, a lack of information makes it difficult to estimate the size of the problem and, then, to track how antimicrobial resistance and multi-drug resistance is changing over time in these and other low- and middle-income countries.

Now, by integrating routinely collected data from a range of databases, Lim, Takahashi et al. estimate that around an extra 19,000 deaths are caused by multi-drug resistant bacteria in Thailand each year. Thailand has a population of about 70 million, and so, per capita, this estimate is about 3 to 5 times larger than those for the United States and European Union (which have a populations of about 300 million and 500 million, respectively). Lim, Takahashi et al. also show that more of the bacteria collected from patients are resistant to multiple antimicrobial drugs and that the burden of antimicrobial resistance in Thailand is worsening over time.

These findings suggest that more studies with a systematic approach need to be done in other low- and middle-income countries, especially in countries where microbiological laboratories are readily available and routinely used. Further work is also needed to identify where resources and attentions are most needed to effectively fight against antimicrobial resistance in low- and middle-income countries.

Attributable mortality, generally defined as the difference in mortality between those with and without the condition of interest, is an important parameter used to estimate the burden of AMR. In the US, it is estimated that mortality from infection attributable to AMR is 6.5%, (*Roberts et al., 2009*) leading to an estimate of 23,000 deaths attributable to AMR each year (*Center for Disease Controls and Prevention and U.S. Department of Health and Human Services, 2013*). In the European Union, it is estimated that the number of deaths attributable to selected antibiotic-resistant bacteria is about 25,000 each year (*European Centre for Disease Prevention and Control and European Medicines Agency, 2009*). There is limited information on mortality attributable to AMR in LMICs. The mortality attributable to ventilator-associated pneumonia in ICUs in Colombia, Peru, and Argentina is estimated to be 17%, 25%, and 35%, respectively, and is associated with a high percentage of AMR organisms (*Moreno et al., 2006*; *Cuellar et al., 2008*; *Rosenthal et al., 2003*). The mortality attributable to ESBL and methicillin-resistance *Staphylococcus aureus* (MRSA) is estimated to be 27% and 34% in neonatal sepsis in Tanzania, respectively, (*Kayange et al., 2010*) which has been used to postulate an estimate that 58,319 deaths could be attributable to ESBL and MRSA in India alone (*Laxminarayan et al., 2013*). In an effort to harmonize the surveillance systems of AMR, a joint initiative between the European Centre for Disease Prevention and Control (ECDC) and the Centres for Disease Prevention and Control (CDC) have developed standard definitions of multi-drug-resistance (MDR) (*Magiorakos et al., 2012*).

We recently combined large data sets from multiple sources including microbiology databases, hospital admission databases, and the national death registry from a sample of ten public hospitals in northeast Thailand from 2004 to 2010 (*Kanoksil et al., 2013*; *Hongsuwan et al., 2014*). We defined community-acquired bacteraemia (CAB) as the isolation of a pathogenic bacterium from blood taken in the first 2 days of admission and without a hospital stay in the 30 days prior to admission, healthcare-associated bacteraemia (HCAB) as the isolation of a pathogenic bacterium from blood taken in the first 2 days of admission and with a hospital stay within 30 days prior to the admission, and hospital-acquired bacteraemia (HAB) as the isolation of a pathogenic bacterium from blood taken after the first 2 days of admission (*Kanoksil et al., 2013*; *Hongsuwan et al., 2014*). We reported an increase in the incidence of CAB, HCAB and HAB over the study period, and that

bacteraemia was associated with high case fatality rates (37.5%, 41.8% and 45.5%, respectively) (*Kanoksil et al., 2013*; *Hongsuwan et al., 2014*). Here, we apply ECDC/CDC standard definitions of MDR to this large data set to evaluate the prevalence, trends, and mortality attributable to MDR bacteria isolated from the blood. We then estimate the number of deaths attributable to MDR in Thailand nationwide.

## Results

We contacted all 20 provincial hospitals in Northeast Thailand to participate in the study. All provincial hospitals were equipped with all basic medical specialties and intensive care units (ICUs). Agreement was obtained from 15 (75%) hospitals, of which ten had hospital databases and microbiological laboratory databases as electronic files in a readily accessible format (*Kanoksil et al., 2013*; *Hongsuwan et al., 2014*). Of these ten hospitals, nine had databases of antimicrobial susceptibility testing results as electronic files for the study (*Figure 1*). The median bed number for the nine hospitals included in the analysis was 450 beds (range 300 to 1000 beds). Of these, three had data available for the period 2004–2010, two between 2007 and 2010, three between 2008 and 2010 and one between 2009 and 2010. Overall, 1,803,506 admission records from 1,255,571 patients were evaluated. A total of 20,803 (1.2%) admission records had at least one blood culture positive for pathogenic organisms during admission. Of 10,022 patients with first episodes of bacteraemia caused by *S. aureus*, *Enterococcus* spp, *E. coli*, *K. pneumoniae*, *P. aeruginosa* and *Acinetobacter* spp., 226 patients (2%) were excluded because the causative organisms were tested for susceptibility to fewer than three antimicrobial categories. Therefore, a total of 9796 first episodes of bacteraemia caused by *S. aureus* (n = 1881), *Enterococcus* spp (n = 342), *E. coli* (n = 4279), *K. pneumoniae* (n = 1661), *P aeruginosa* (n = 568), and *Acinetobacter* spp. (n = 1065) were evaluated in the analysis. The proportion of bacteria being MDR was highest in HAB and lowest in CAB for all organisms (all p<0.001 except for *Enterococcus* spp., *Table 1*).

### Staphylococcus aureus

Of CAB, HCAB and HAB caused by *S. aureus*, 8%, 28%, and 50% were caused by MDR *S. aureus*, respectively (p<0.001). Almost all MDR *S. aureus* were MRSA (92% [357/389], *Table 2*). We did not observe a trend in the proportion of *S. aureus* bacteraemia being caused by MRSA (*Figure 2*). Vancomycin non-susceptible *S. aureus* was found in <1% of tested isolates (6/1380).

### *Enterococcus* species

MDR *Enterococcus* spp. were not found in CAB (0/176) and HCAB (0/49), while 3% (4/117) of *Enterococcus* spp. causing HAB were MDR. Of CAB caused by *Enterococcus* spp., 15% (20/134) and 23% (35/153) was non-susceptible to ampicillin and gentamicin, respectively (*Table 3*), while 42% (34/81) and 62% (63/101) of HAB caused by *Enterococcus* spp. were non-susceptible to those agents, respectively (both p<0.001). Vancomycin non-susceptible *Enterococcus* spp. was found in 4% of tested isolates (15/338).

### Escherichia coli

Of CAB, HCAB and HAB caused by *E. coli*, 35%, 58% and 63% were caused by MDR *E. coli*, respectively (p<0.001). Of *E. coli* causing CAB, 79% (2246/2843), 16% (501/3076), 24% (728/3000), 58% (1738/3007), and 17% (559/3346) were non-susceptible to commonly-used antimicrobials for community-acquired infections such as ampicillin, cefotaxime, ciprofloxacin, trimethoprim-sulphamethoxazole, and gentamicin, respectively (*Table 4*). From 2004 to 2010, the proportions of community-acquired *E. coli* bacteraemia being caused by *E. coli* non-susceptible to extended-spectrum cephalosporins rose from 5% (9/169) to 23% (186/815) (p=0.04) (*Figure 3*). The proportions of healthcare-associated and hospital-acquired *E. coli* bacteraemia being caused by *E. coli* non-susceptible to extended-spectrum cephalosporins were high (44% [204/465] and 52% [190/368], respectively), but a significant trend over time was not observed (p=0.18 and p=0.63, respectively). Carbapenem non-susceptible *E. coli* was found in <1% of tested isolates (12/3838).

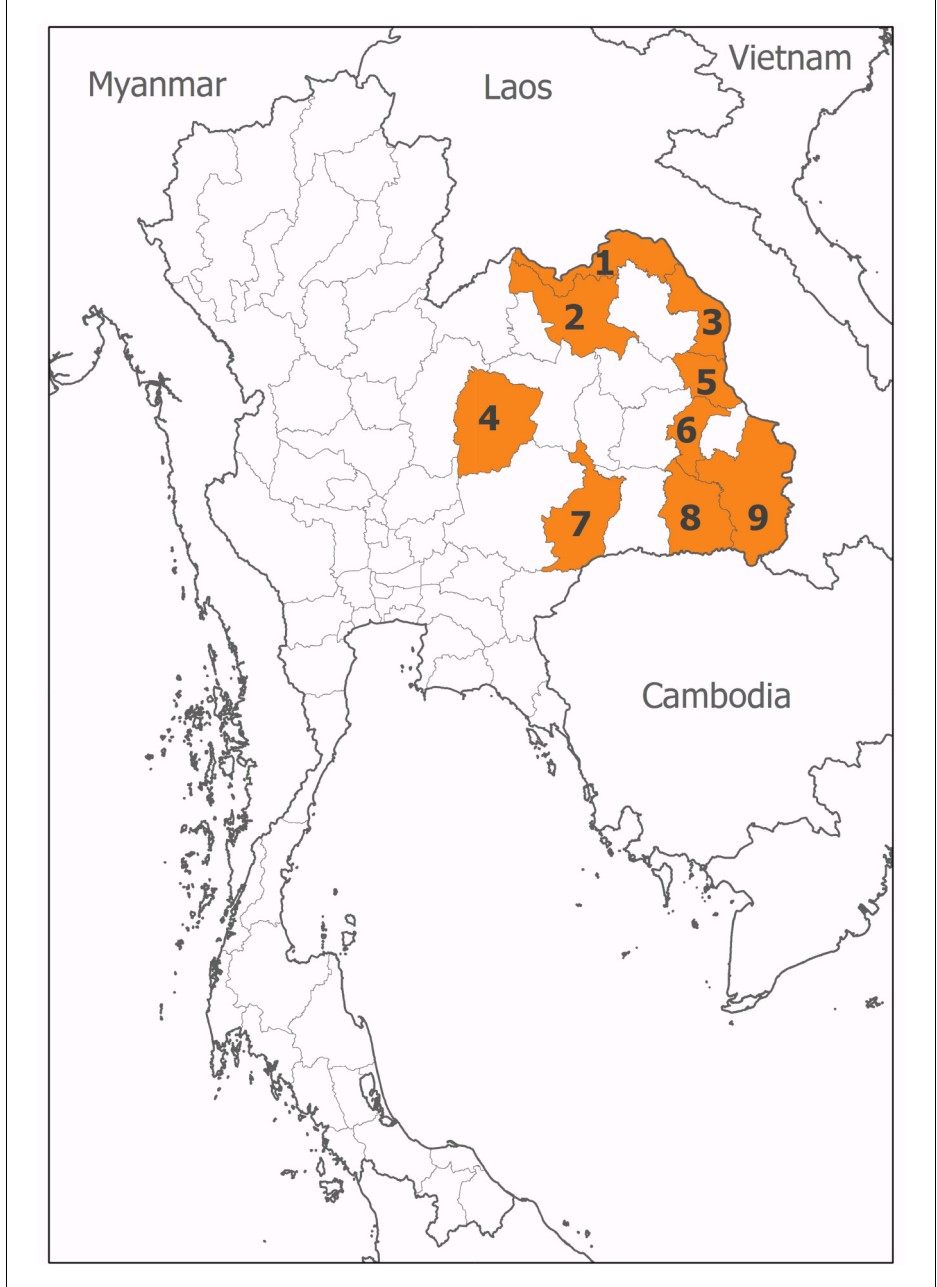

**Figure 1.** Location of participating hospitals. These were situated in (1) Nong Khai, (2) Udon Thani, (3) Nakhon Phanom, (4) Chaiyaphum, (5) Mukdahan, (6) Yasothon, (7) Burirum, (8) Sisaket, and (9) Ubon Ratchathani provinces.

## Klebsiella pneumoniae

Of CAB, HCAB and HAB caused by *K. pneumoniae*, 14%, 36%, and 66% were caused by MDR *K. pneumoniae*, respectively (p<0.001). Of *K. pneumoniae* causing CAB, 16% (146/902), 16% (143/894), 23% (198/876), and 9% (94/999) were non-susceptible to cefotaxime, ciprofloxacin, trimethoprim-sulphamethoxazole and gentamicin, respectively (*Table 5*). From 2004 to 2010, the proportions of community-acquired *K. pneumoniae* bacteraemia being caused by *K. pneumoniae* non-susceptible to extended-spectrum cephalosporins rose from 12% (6/50) to 24% (64/263) (p=0.04) (*Figure 4*). The proportions of healthcare-associated and hospital-acquired *K. pneumoniae* bacteraemia being caused by *K. pneumoniae* non-susceptible to extended-spectrum cephalosporins were also high

**Table 1.** Proportions of bacteraemias being caused by multidrug-resistant (MDR) variants of those bacteria.

| Pathogens | Community-acquired bacteraemia (CAB) | Healthcare-associated bacteraemia (HCAB) | Hospital-acquired bacteraemia (HAB) | p values |
|---|---|---|---|---|
| MDR *Staphylococcus aureus* | 94/1176 (8%) | 73/259 (28%) | 222/446 (50%) | <0.001 |
| MDR *Enterococcus* spp | 0/176 (0%) | 0/49 (0%) | 4/117 (3%) | 0.02 |
| MDR *Escherichia coli* | 1177/3382 (35%) | 288/494 (58%) | 252/403 (63%) | <0.001 |
| MDR *Klebsiella pneumoniae* | 146/1010 (14%) | 71/196 (36%) | 301/455 (66%) | <0.001 |
| MDR *Pseudomonas aeruginosa* | 13/286 (5%) | 10/103 (10%) | 45/179 (25%) | <0.001 |
| MDR *Acinetobacter* spp | 125/449 (28%) | 58/115 (50%) | 374/501 (75%) | <0.001 |

NOTE: CAB was defined as the isolation of a pathogenic bacterium from blood taken in the first 2 days of admission and without a hospital stay in the 30 days prior to admission. HCAB was defined as the isolation of a pathogenic bacterium from blood taken in the first 2 days of admission and with a hospital stay within 30 days prior to the admission. HAB was defined as the isolation of a pathogenic bacterium from blood taken after the first 2 days of admission.

**Table 2.** Antibiogram of *S. aureus* causing bacteraemia in Northeast Thailand.

| Antibiotic category | Antibiotic agents | CAB (n = 1176 patients) | HCAB (n = 259 patients) | HAB (n = 446 patients) | p values |
|---|---|---|---|---|---|
| Aminoglycosides | Gentamicin | 24/484 (5%) | 16/84 (19%) | 66/151 (44%) | <0.001 |
| Ansamycins | Rifampin | 2/129 (2%) | 1/19 (5%) | 0/38 (0%) | 0.37 |
| Anti-MRSA cephalosporins | Ceftaroline | NA | NA | NA | - |
| Cefamycins | Oxacillin * | 80/1145 (7%) | 67/247 (27%) | 210/441 (48%) | <0.001 |
| Fluoroquinolones | Ciprofloxacin | 3/45 (7%) | 2/8 (25%) | 4/10 (40%) | 0.01 |
| | Moxifloxacin | NA | NA | NA | - |
| Folate pathway inhibitors | Trimethoprim-sulphamethoxazole | 99/1139 (9%) | 57/251 (23%) | 185/438 (42%) | <0.001 |
| Fucidanes | Fusidic acid | 33/618 (5%) | 4/170 (2%) | 12/291 (4%) | 0.26 |
| Glycopeptides | Vancomycin † | 4/833 (0.5%) | 0/190 (0%) | 2/357 (1%) | 0.86 |
| | Teicoplanin | 2/66 (3%) | 1/17 (6%) | 0/17 (0%) | 0.72 |
| | Telavancin | NA | NA | NA | - |
| Glycylcyclines | Tigecycline | NA | NA | NA | - |
| Lincosamides | Clindamycin | 118/1147 (10%) | 77/251 (31%) | 202/438 (46%) | <0.001 |
| Lipopeptides | Daptomycin | NA | NA | NA | - |
| Macrolides | Erythromycin | 138/1116 (12%) | 76/240 (32%) | 222/429 (52%) | <0.001 |
| Oxazolidinones | Linezolid | 0/81 (0%) | 0/16 (0%) | 0/32 (0%) | - |
| Phenicols | Chloramphenicol | 6/86 (7%) | 4/24 (17%) | 2/14 (14%) | 0.21 |
| Phosphonic acids | Fosfomycin | 14/361 (4%) | 10/66 (15%) | 24/141 (17%) | <0.001 |
| Streptogramins | Quinupristin-dalfopristin | NA | NA | NA | - |
| Tetracyclines | Tetracycline | NA | NA | NA | - |
| | Doxycycline | NA | NA | NA | - |
| | Minocycline | NA | NA | NA | - |
| MDR | | 94/1176 (8%) | 73/259 (28%) | 222/446 (50%) | <0.001 |

NOTE: Data are number of isolates demonstrating non-susceptible to the antimicrobial over the total number of isolates tested (%). CAB = Community-acquired bacteraemia, HCAB = Healthcare-associated bacteraemia, HAB = Hospital-acquired bacteraemia, and NA = Not available. The first isolate of each patient was used. MDR (one or more of these have to apply): (i) an MRSA is always considered MDR by virtue of being an MRSA (ii) non-susceptible to ≥1 agent in ≥3 antimicrobial categories.

* Defined by using a 30 μg cefoxitin disc and an inhibition zone diameter of <21 mm.

† Defined by using a 30 μg vancomycin disc and an inhibition zone diameter of <15 mm.

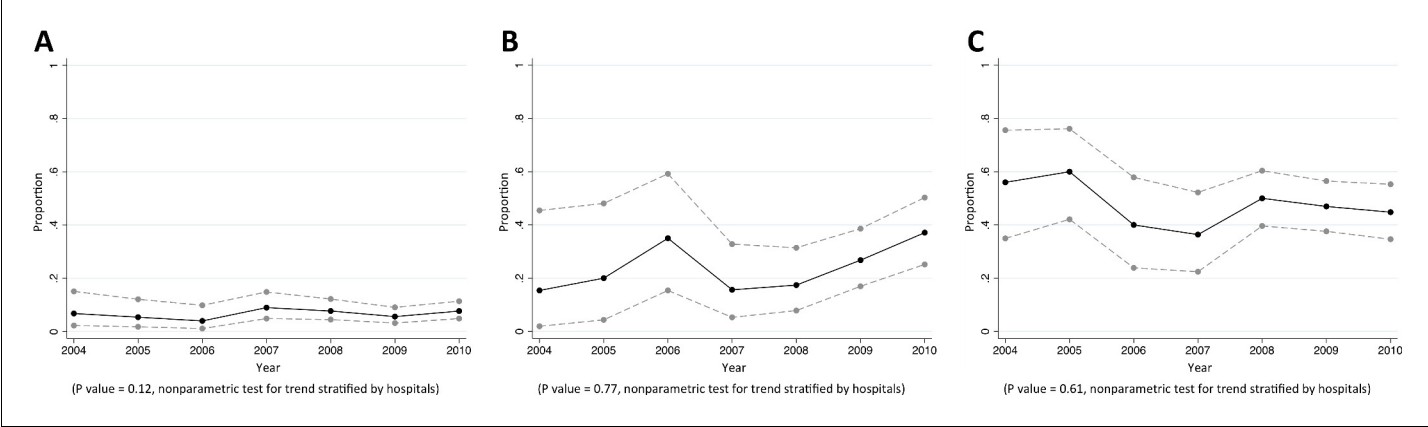

**Figure 2.** Trends in proportions of *Staphylococcus aureus* bacteraemia being caused by MRSA in Northeast Thailand. (**A**) community-acquired, (**B**) healthcare-associated and (**C**) hospital-acquired *Staphylococcus aureus* bacteraemia.

**Table 3.** Antibiogram of *Enterococcus* spp. causing bacteraemia in Northeast Thailand.

| Antibiotic category | Antibiotic agents | CAB (n = 176 patients) | HCAB (n = 49 patients) | HAB (n = 117 patients) | p values |
|---|---|---|---|---|---|
| Aminoglycosides | Gentamicin (high level) | 35/153 (23%) | 24/45 (53%) | 63/101 (62%) | <0.001 |
| Streptomycin | Streptomycin (high level) | NA | NA | NA | - |
| Carbapenems* | Imipenem | NA | NA | NA | - |
| | Meropenem | 1/1 (100%) | NA | 3/5 (60%) | >0.99 |
| | Doripenem | NA | NA | NA | - |
| Fluoroquinolones | Ciprofloxacin | 37/44 (84%) | 9/10 (90%) | 31/37 (84%) | >0.99 |
| | Levofloxacin | 5/18 (28%) | 1/6 (17%) | 11/15 (73%) | 0.01 |
| | Moxifloxacin | NA | NA | NA | - |
| Glycopeptides | Vancomycin | 9/176 (5%) | 0/49 (0%) | 6/113 (5%) | 0.27 |
| | Teicoplanin | 0/11 (0%) | 0/4 (0%) | 0/10 (0%) | - |
| Glycylcyclines | Tigecycline | NA | NA | NA | - |
| Lipopeptides | Daptomycin | NA | NA | NA | - |
| Oxazolidinones | Linezolid | 0/8 (0%) | 0/2 (0%) | 0/4 (0%) | - |
| Penicillins | Ampicillin | 20/134 (15%) | 6/37 (16%) | 34/81 (42%) | <0.001 |
| Streptogramins* | Quinupristin-dalfopristin | NA | NA | NA | - |
| Tetracycline | Doxycycline | NA | NA | NA | - |
| | Minocycline | NA | NA | NA | - |
| MDR | | 0/176 (0%) | 0/49 (0%) | 4/117 (3%) | 0.02 |

NOTE: Data are number of isolates demonstrating non-susceptible to the antimicrobial over the total number of isolates tested (%). CAB = Community-acquired bacteraemia, HCAB = Healthcare-associated bacteraemia, HAB = Hospital-acquired bacteraemia, and NA = Not available. The first isolate of each patient was used. MDR: non-susceptible to ≥1 agent in ≥3 antimicrobial categories.

*Intrinsic resistance in *E. faecium* against carbapenems and in *E. faecalis* against streptogramins. When a species has intrinsic resistance to an antimicrobial category, that category is removed prior to applying the criteria for the MDR definition and is not counted when calculating the number of categories to which the bacterial isolate is non-susceptible.

**Table 4.** Antibiogram of *E. coli* causing bacteraemia in Northeast Thailand.

| Antibiotic category | Antibiotic agents | CAB (n = 3382 patients) | HCAB (n = 494 patients) | HAB (n = 403 patients) | p values |
|---|---|---|---|---|---|
| Aminoglycosides | Gentamicin | 559/3346 (17%) | 166/484 (34%) | 178/398 (45%) | <0.001 |
| | Tobramycin | NA | NA | NA | - |
| | Amikacin | 72/2685 (3%) | 26/397 (7%) | 32/326 (10%) | <0.001 |
| | Netilmicin | 68/1394 (5%) | 25/259 (10%) | 42/254 (17%) | <0.001 |
| Anti-MRSA cephalosporins | Ceftaroline | NA | NA | NA | - |
| Antipseudomonal penicillins + β lactamase inhibitors | Ticarcillin-clauvanic acid | NA | NA | NA | - |
| | Piperacillin-tazobactam | 23/511 (5%) | 10/103 (10%) | 15/89 (17%) | <0.001 |
| Carbapenems | Ertapenem | 4/1325 (<1%) | 1/235 (<1%) | 4/205 (2%) | 0.02 |
| | Imipenem | 3/2449 (<1%) | 0/386 (0%) | 3/344 (1%) | 0.04 |
| | Meropenem | 0/1988 (0%) | 1/314 (<1%) | 1/244 (<1%) | 0.05 |
| Non-extended spectrum cephalosporins | Cefazolin | 468/1095 (43%) | 115/174 (66%) | 80/102 (78%) | <0.001 |
| | Cefuroxime | 219/1438 (15%) | 96/226 (42%) | 102/202 (50%) | <0.001 |
| Extended-spectrum cephalosporins | Cefotaxime | 501/3076 (16%) | 199/455 (44%) | 185/361 (51%) | <0.001 |
| | Ceftazidime | 392/3020 (13%) | 165/446 (37%) | 164/351 (47%) | <0.001 |
| | Cefepime | 30/293 (10%) | 12/42 (29%) | 18/53 (34%) | <0.001 |
| Cephamycins | Cefoxitin | 36/1200 (3%) | 16/215 (7%) | 16/195 (8%) | <0.001 |
| | Cefotetan | NA | NA | NA | - |
| Fluoroquinolones | Ciprofloxacin | 728/3000 (24%) | 221/452 (49%) | 171/384 (45%) | <0.001 |
| Folate pathway inhibitors | Trimethoprim-sulphamethoxazole | 1738/3007 (58%) | 294/442 (67%) | 225/350 (64%) | <0.001 |
| Glycylcyclines | Tigecycline | 0/7 (0%) | NA | 0/1 (0%) | - |
| Monobactams | Aztreonam | NA | NA | NA | - |
| Penicillins | Ampicillin | 2246/2843 (79%) | 371/420 (88%) | 301/342 (88%) | <0.001 |
| Penicillins + β lactamase inhibitors | Amoxicillin-clavulanic acid | 790/3074 (26%) | 191/463 (41%) | 158/373 (42%) | <0.001 |
| | Ampicillin-sulbactam | 83/296 (28%) | 18/48 (38%) | 12/25 (48%) | 0.06 |
| Phenicols | Chloramphenicol | 14/63 (22%) | 1/4 (25%) | 3/5 (60%) | 0.14 |
| Phosphonic acids | Fosfomycin | NA | NA | NA | - |
| Polymyxins | Colistin* | 2/34 (6%) | 0/6 (0%) | 1/6 (17%) | 0.61 |
| MDR | | 1177/3382 (35%) | 288/494 (58%) | 252/403 (63%) | <0.001 |

NOTE: Data are number of isolates demonstrating non-susceptible to the antimicrobial over the total number of isolates tested (%). CAB = Community-acquired bacteraemia, HCAB = Healthcare-associated bacteraemia, HAB = Hospital-acquired bacteraemia, and NA = Not available. The first isolate of each patient was used. MDR: non-susceptible to ≥1 agent in ≥3 antimicrobial categories.

*Defined by using an inhibition zone of <11 mm.

(40% [71/177] and 71% [304/431], respectively), but a significant trend over time was not observed (p=0.16 and p=0.35, respectively). Carbapenem non-susceptible *K. pneumoniae* was found in <1% of tested isolates (11/1555).

## Pseudomonas aeruginosa

Of CAB, HCAB and HAB caused by *P. aeruginosa*, 5%, 10%, and 25% were caused by MDR *P. aeruginosa*, respectively (p<0.001). Of *P. aeruginosa* causing HAB, 38% (68/179), 27% (48/177), 23% (39/169) and 26% (42/164) were non-susceptible to commonly-used antimicrobials for HAI such as ceftazidime, amikacin, ciprofloxacin and carbapenems, respectively (*Table 6*). We did not observe a trend in the proportions of *P. aeruginosa* being caused by *P. aeruginosa* that were non-susceptible to any specific antibiotic group (*Figure 5*).

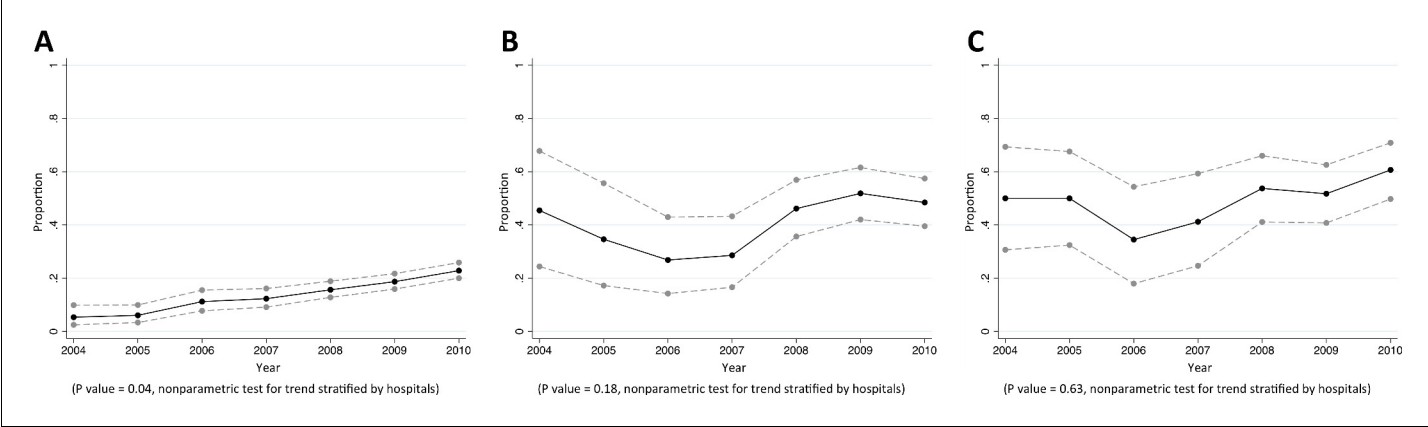

**Figure 3.** Trends in proportions of *Escherichia coli* bacteraemia being caused by *E. coli* non-susceptible to extended-spectrum cephalosporins in Northeast Thailand. (**A**) community-acquired, (**B**) healthcare-associated and (**C**) hospital-acquired *E. coli* bacteraemia.

## *Acinetobacter* species

Of CAB, HCAB and HAB caused by *Acinetobacter* spp., 28%, 50%, and 75% were caused by MDR *Acinetobacter* spp., respectively (p<0.001). Of *Acinetobacter* spp. causing HAB, 75% (377/500), 63% (310/495), 67% (322/481) and 64% (315/490) were non-susceptible to ceftazidime, amikacin, cipro-floxacin and carbapenems, respectively (*Table 7*). There was borderline evidence that the proportion of hospital-acquired *Acinetobacter spp.* bacteraemia being caused by *Acinetobacter spp.* non-sus-ceptible to carbapenem rose from 49% (19/39) in 2004 to 65% (70/108) in 2010 (p=0.10) (*Figure 6*). Non-susceptibility to colistin was observed in 3% of tested isolates (2/63).

## Mortality attributable to MDR

The 30-day mortality in patients with CAB, HCAB and HAB caused by MDR bacteria was 35% (549/1555), 49% (247/500), and 53% (640/1198), compared with 32% (1595/4924), 37% (264/716), and 42% (383/903) in CAB, HCAB, and HAB caused by non-MDR bacteria, respectively (*Figure 7*). In the final multivariable logistic regression model, gender, age group, year of admission and time to bac-teraemia (for HAB) were included (*Supplementary file 2*).

If excess mortality in patients infected with MDR bacteria after adjusting for confounding factors in the final multivariable model is assumed to be caused by MDR, the mortality attributable to MDR was 7% (95%CI 4% to 10%, p<0.001) in CAB, 15% (95%CI 5% to 24%, p<0.001) in HCAB and 15%

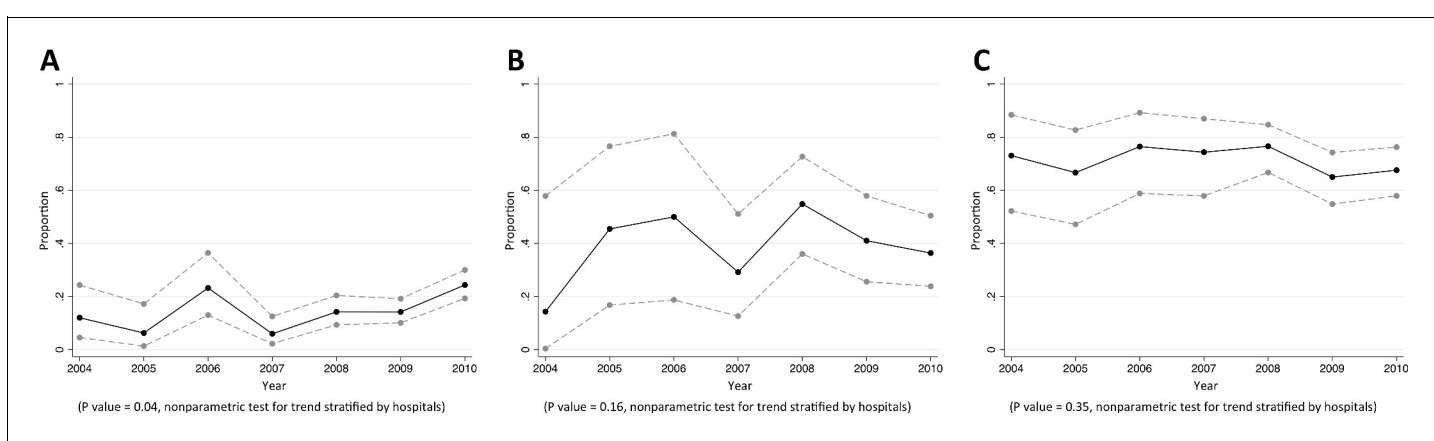

**Figure 4.** Trends in proportions of *Klebsiella pneumoniae* bacteraemia being caused by *K. pneumoniae* non-susceptible to extended-spectrum cephalosporins in Northeast Thailand. (**A**) community-acquired, (**B**) healthcare-associated and (**C**) hospital-acquired *K. pneumoniae* bacteraemia.

**Table 5.** Antibiogram of *K. pneumoniae* causing bacteraemia in Northeast Thailand.

| Antibiotic category | Antibiotic agents | CAB (n = 1010 patients) | HCAB (n = 196 patients) | HAB (n = 455 patients) | p values |
|---|---|---|---|---|---|
| Aminoglycosides | Gentamicin | 94/999 (9%) | 53/193 (27%) | 265/444 (60%) | <0.001 |
| | Tobramycin | NA | NA | NA | - |
| | Amikacin | 17/815 (2%) | 12/157 (8%) | 109/398 (27%) | <0.001 |
| | Netilmicin | 20/450 (4%) | 23/112 (21%) | 124/320 (39%) | <0.001 |
| Anti-MRSA cephalosporins | Ceftaroline | NA | NA | NA | - |
| Antipseudomonal penicillins + β lactamase inhibitors | Ticarcillin-clauvanic acid | NA | NA | NA | - |
| | Piperacillin-tazobactam | 24/166 (14%) | 14/32 (44%) | 73/121 (60%) | <0.001 |
| Carbapenems | Ertapenem | 2/432 (0%) | 1/100 (1%) | 5/264 (2%) | 0.17 |
| | Imipenem | 1/778 (0%) | 1/164 (1%) | 2/408 (0%) | 0.24 |
| | Meropenem | 0/583 (0%) | 1/113 (1%) | 2/317 (1%) | 0.10 |
| Non-extended spectrum cephalosporins | Cefazolin | 76/319 (24%) | 30/60 (50%) | 101/127 (80%) | <0.001 |
| | Cefuroxime | 81/478 (17%) | 35/98 (36%) | 161/231 (70%) | <0.001 |
| Extended-spectrum cephalosporins | Cefotaxime | 146/902 (16%) | 71/173 (41%) | 298/424 (70%) | <0.001 |
| | Ceftazidime | 124/927 (13%) | 63/176 (36%) | 295/430 (69%) | <0.001 |
| | Cefepime | 5/100 (5%) | 8/22 (36%) | 25/51 (49%) | <0.001 |
| Cephamycins | Cefoxitin | 15/396 (4%) | 10/95 (11%) | 14/230 (6%) | 0.03 |
| | Cefotetan | NA | NA | NA | - |
| Fluoroquinolones | Ciprofloxacin | 143/894 (16%) | 66/176 (38%) | 187/430 (43%) | <0.001 |
| Folate pathway inhibitors | Trimethoprim-sulphamethoxazole | 198/876 (23%) | 69/171 (40%) | 219/407 (54%) | <0.001 |
| Glycylcyclines | Tigecycline | NA | NA | NA | - |
| Monobactams | Aztreonam | NA | NA | NA | - |
| Penicillins + β lactamase inhibitors | Amoxicillin-clavulanic acid | 131/945 (14%) | 68/183 (37%) | 291/443 (66%) | <0.001 |
| | Ampicillin-sulbactam | 20/105 (19%) | 9/17 (53%) | 23/38 (61%) | <0.001 |
| Phenicols | Chloramphenicol | 4/19 (21%) | 0/2 (0%) | 0/3 (0%) | >0.99 |
| Phosphonic acids | Fosfomycin | NA | NA | NA | - |
| Polymyxins | Colistin * | 0/6 (0%) | 0/2 (0%) | 0/5 (0%) | - |
| MDR | | 146/1010 (14%) | 71/196 (36%) | 301/455 (66%) | <0.001 |

NOTE: Data are number of isolates demonstrating non-susceptible to the antimicrobial over the total number of isolates tested (%). CAB = Community-acquired bacteraemia, HCAB = Healthcare-associated bacteraemia, HAB = Hospital-acquired bacteraemia, and NA = Not available. The first isolate of each patient was used. MDR: non-susceptible to ≥1 agent in ≥3 antimicrobial categories.

* Defined by using an inhibition zone of <11 mm.

(95%CI 2% to 27%, p<0.001) in HAB (*Figure 7*). Heterogeneity between different organisms was clearly observed in HAB (p<0.001), and borderline evidence of heterogeneity was observed in HCAB (p=0.09). The heterogeneity observed in HCAB and HAB was largely caused by MDR *Acinetobacter* spp. (*Figure 7B and C*). Mortality attributed to MDR was highest for hospital-acquired MDR *Acinetobacter* bacteraemia (41%).

Using our estimated mortality attributed to MDR bacteraemia (*Figure 7C*) and national statistics of HAI caused by MDR bacteria, we further estimated that 19,122 of 45,209 (43%) deaths in patients with HAI due to MDR bacteria in Thailand in 2010 represented excess mortality caused by MDR (*Table 8*). All parameters used to estimate the number of excess deaths in Thailand are shown in *Supplementary file 2*.

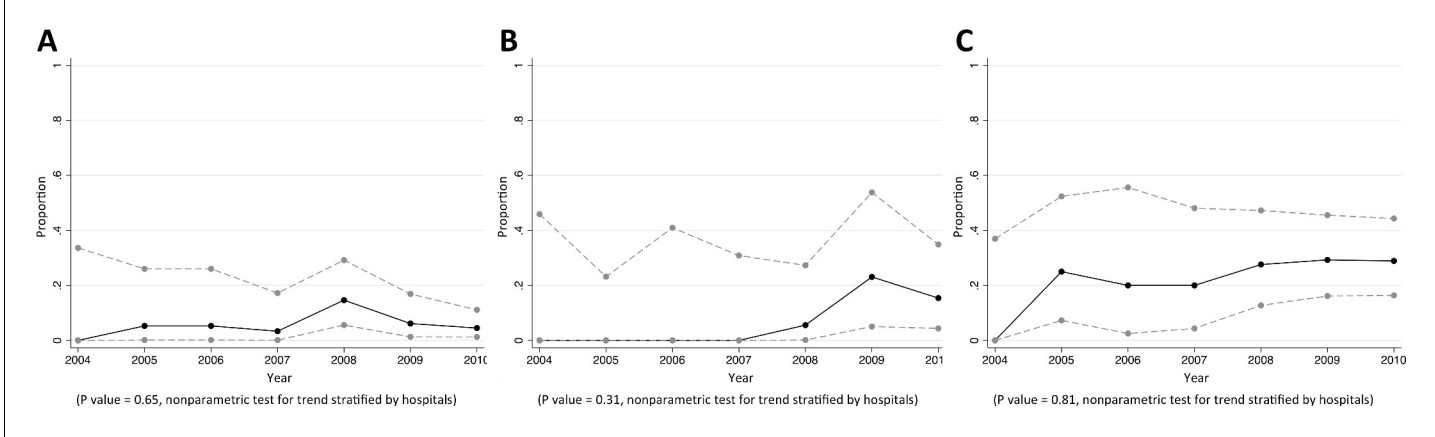

**Figure 5.** Trends in proportions of *Pseudomonas aeruginosa* bacteraemia being caused by *P. aeruginosa* non-susceptible to carbapenem in Northeast Thailand. (**A**) community-acquired, (**B**) healthcare-associated and (**C**) hospital-acquired *Pseudomonas aeruginosa* bacteraemia.

## Discussion

This study presents detailed antimicrobial susceptibility data on common pathogenic bacteria, the association of MDR with infection acquisition (community-acquired, healthcare-associated and hospital-acquired), and excess mortality from MDR in a developing country. Our estimate of excess deaths caused by MDR in HAI patients in Thailand (19,122 deaths per year in a country of about 66 million population in 2010) is large compared to those estimated in USA (23,000 death per year in a country

**Table 6.** Antibiogram of *P. aeruginosa* causing bacteraemia in Northeast Thailand.

| Antibiotic category | Antibiotic agents | CAB (n = 286 patients) | HCAB (n = 103 patients) | HAB (n = 179 patients) | p values |
|---|---|---|---|---|---|
| Aminoglycosides | Gentamicin | 29/235 (12%) | 13/88 (15%) | 60/140 (43%) | <0.001 |
| | Tobramycin | NA | NA | NA | - |
| | Amikacin | 27/284 (10%) | 13/100 (13%) | 48/177 (27%) | <0.001 |
| | Netilmicin | 8/155 (5%) | 5/67 (7%) | 34/120 (28%) | <0.001 |
| Antipseudomonal carbapenems | Imipenem | 14/238 (6%) | 6/86 (7%) | 37/154 (24%) | <0.001 |
| | Meropenem | 9/163 (6%) | 8/73 (11%) | 24/125 (19%) | 0.001 |
| | Doripenem | 2/17 (12%) | 0/3 (0%) | 2/2 (100%) | 0.04 |
| Antipseudomonal cephalosporins | Ceftazidime | 29/280 (10%) | 16/103 (16%) | 68/179 (38%) | <0.001 |
| | Cefepime | 2/36 (6%) | 2/18 (11%) | 10/28 (36%) | 0.01 |
| Antipseudomonal fluoroquinolones | Ciprofloxacin | 24/275 (9%) | 12/101 (12%) | 39/169 (23%) | <0.001 |
| | Levofloxacin | 0/1 (0%) | 1/1 (100%) | 1/1 (100%) | >0.99 |
| Antipseudomonal penicillins + β lactamase inhibitors | Ticarcillin-clauvanic acid | NA | NA | NA | - |
| | Piperacillin-tazobactam | 8/85 (9%) | 6/38 (16%) | 8/46 (17%) | 0.37 |
| Monobactams | Aztreonam | NA | NA | NA | - |
| Phosphonic acids | Fosfomycin | 1/1 (100%) | NA | NA | - |
| Polymyxins | Colistin | 0/7 (0%) | 0/3 (0%) | 1/7 (14%) | >0.99 |
| | Polymyxin B | NA | NA | NA | - |
| MDR | | 13/286 (5%) | 10/103 (10%) | 45/179 (25%) | <0.001 |

NOTE: Data are number of isolates demonstrating non-susceptible to the antimicrobial over the total number of isolates tested (%). CAB = Community-acquired bacteraemia, HCAB = Healthcare-associated bacteraemia, HAB = Hospital-acquired bacteraemia, and NA = Not available. The first isolate of each patient was used. MDR: non-susceptible to ≥1 agent in ≥3 antimicrobial categories.

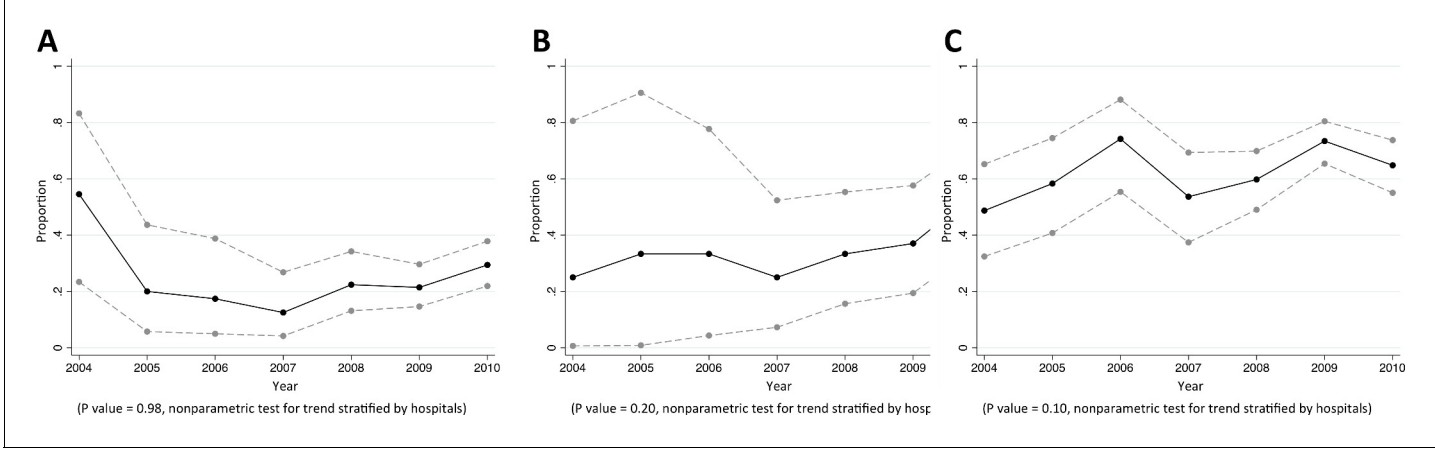

**Figure 6.** Trends in proportions of *Acinetobacter* spp bacteraemia being caused by *Acinetobacter* spp non-susceptible to carbapenem in Northeast Thailand. (**A**) community-acquired, (**B**) healthcare-associated and (**C**) hospital-acquired *Acinetobacter* spp bacteraemia.

**Table 7.** Antibiogram of *Acinetobacter* spp. causing bacteraemia in Northeast Thailand.

| Antibiotic category | Antibiotic agents | CAB (n = 449 patients) | HCAB (n = 115 patients) | HAB (n = 501 patients) | p values |
|---|---|---|---|---|---|
| Aminoglycosides | Gentamicin | 112/390 (29%) | 45/105 (43%) | 310/455 (68%) | <0.001 |
| | Tobramycin | NA | NA | NA | - |
| | Amikacin | 123/442 (28%) | 45/112 (40%) | 310/495 (63%) | <0.001 |
| | Netilmicin | 44/203 (22%) | 24/64 (38%) | 224/381 (59%) | <0.001 |
| Antipseudomonal carbapenems | Imipenem | 87/397 (22%) | 37/102 (36%) | 293/459 (64%) | <0.001 |
| | Meropenem | 65/284 (23%) | 32/81 (40%) | 229/348 (66%) | <0.001 |
| | Doripenem | 16/45 (36%) | 9/10 (90%) | 6/7 (86%) | 0.001 |
| Antipseudomonal fluoroquinolones | Ciprofloxacin | 84/413 (20%) | 53/106 (50%) | 322/481 (67%) | <0.001 |
| | Levofloxacin | 2/5 (40%) | 2/2 (100%) | 8/9 (89%) | 0.11 |
| Antipseudomonal penicillins + β lactamase inhibitors | Ticarcillin- clauvanic acid | NA | NA | NA | - |
| | Piperacillin-tazobactam | 22/98 (22%) | 13/28 (46%) | 74/106 (70%) | <0.001 |
| Extended-spectrum cephalosporins | Cefotaxime | 242/291 (83%) | 89/94 (95%) | 407/420 (97%) | <0.001 |
| | Ceftazidime | 133/448 (30%) | 61/114 (54%) | 377/500 (75%) | <0.001 |
| | Cefepime | 18/53 (34%) | 10/22 (45%) | 95/133 (71%) | <0.001 |
| Folate pathway inhibitor | Trimethopri-sulphamethoxazole | 119/356 (33%) | 55/99 (56%) | 333/435 (77%) | <0.001 |
| Penicillins + β lactamase inhibitors | Ampicillin-sulbactam | 43/134 (32%) | 16/29 (55%) | 79/115 (69%) | <0.001 |
| Polymyxins | Colistin * | 2/16 (13%) | 0/14 (0%) | 0/33 (0%) | 0.11 |
| | Polymyxin B | NA | NA | NA | - |
| Tetracyclines | Tetracycline | NA | NA | NA | - |
| | Doxycycline | NA | NA | NA | - |
| | Minocycline | NA | NA | NA | - |
| MDR | | 125/449 (28%) | 58/115 (50%) | 374/501 (75%) | <0.001 |

NOTE: Data are number of isolates demonstrating non-susceptible to the antimicrobial over the total number of isolates tested (%). CAB = Community-acquired bacteraemia, HCAB = Healthcare-associated bacteraemia, HAB = Hospital-acquired bacteraemia, and NA = Not available. The first isolate of each patient was used. MDR: non-susceptible to ≥1 agent in ≥3 antimicrobial categories.
* Defined by using an inhibition zone of <11 mm.

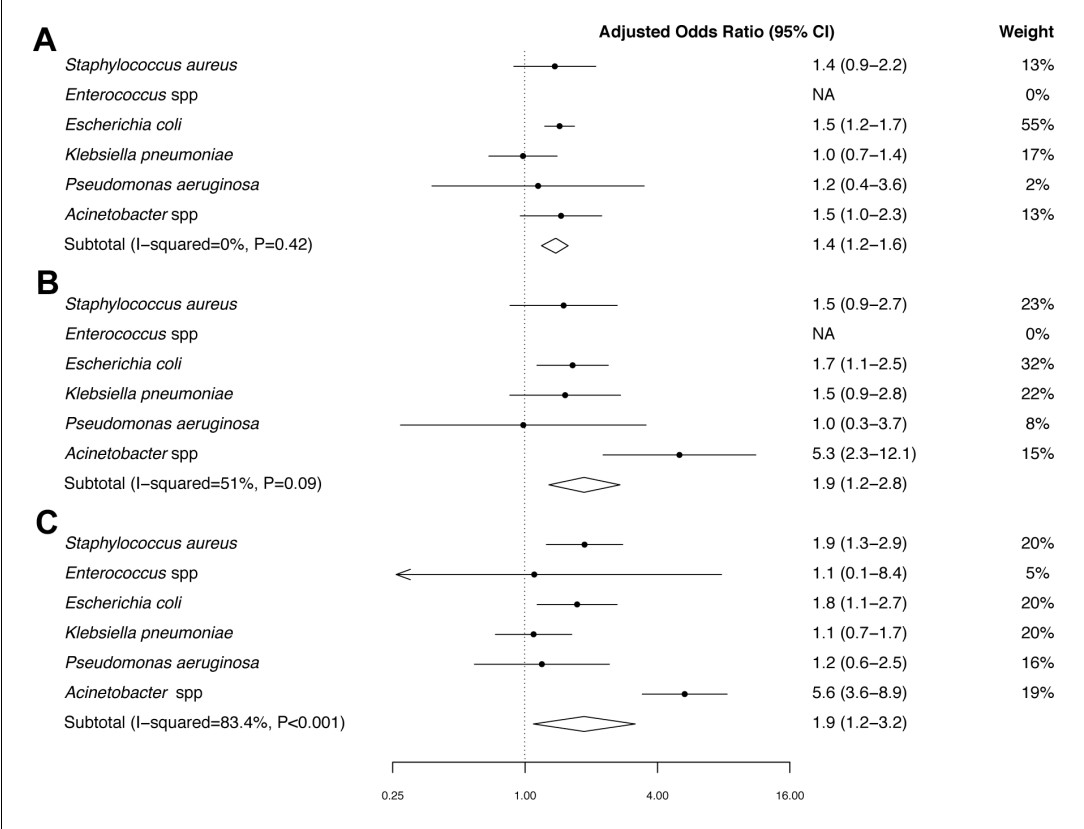

**Figure 7.** Forest plot of mortality in patients with MDR bacteraemia compared with non-MDR bacteraemia in Northeast Thailand. (**A**) Community-acquired bacteraemia. (**B**) Healthcare-associated bacteraemia. (**C**) Hospital-acquired bacteraemia.

The following source data is available for figure 7:

**Source data 1.** Mortality in patients with MDR and non-MDR bacteraemia in Northeast Thailand.

**Table 8.** Estimates of mortality attributable to multidrug-resistance (MDR) in hospital-acquired infection (HAI) in Thailand.

| Pathogens | No of patients* | Estimated mortality (%)[†] | Estimated mortality if the infections were caused by non-MDR organisms (%)[†, ‡] | Estimated excess mortality caused by MDR (%)[†, ‡] |
|---|---|---|---|---|
| MDR *Staphylococcus aureus* | 18,725 | 8262 (44%) | 5463 (29%) | 2799 (15%) |
| MDR *Escherichia coli* | 11,116 | 2163 (19%) | 1566 (14%) | 597 (5%) |
| MDR *Klebsiella pneumoniae* | 15,239 | 5267 (35%) | 4979 (33%) | 288 (2%) |
| MDR *Pseudomonas aeruginosa* | 6118 | 3966 (65%) | 3696 (60%) | 270 (4%) |
| MDR *Acinetobacter* spp | 36,553 | 25,551 (70%) | 10,383 (28%) | 15,168 (41%) |
| Total | 87,751 | 45,209 (52%) | 26,087 (30%) | 19,122 (22%) |

*Cumulative incidence of antimicrobial resistant HAI in Thailand 2010 estimated by **Pumart et al. (2012)**.

[†]All parameters used to estimate the mortality and excess mortality are shown in **Supplementary file 2**.

[‡]Excess mortality caused by MDR (mortality attributable to MDR) was defined as the difference in mortality of patients with MDR infection and their mortality if they were infected with non-MDR infections.

of 316 million population in 2013) (*Center for Disease Controls and Prevention and U.S. Department of Health and Human Services, 2013*) and the European Union (25,000 deaths per year in EU of about 500 million population in 2007) (*European Centre for Disease Prevention and Control and European Medicines Agency, 2009*). Our study highlights the need for public health officials and international health organizations to improve systems to track and reduce the burden of AMR in LMICs. Our estimated mortality for those with MDR HAI (45,209, *Table 2*) is higher than those previously published by Pumart et al. (38,481) (*Pumart et al., 2012*), probably because we used 30-day mortality rather than in-hospital mortality.

*Acinetobacter* spp. is increasingly recognized as an important cause of HAI, (*Munoz-Price and Weinstein, 2008*; *Peleg and Hooper, 2010*) and our study confirms the importance of this species as a leading cause of hospital-acquired MDR infection in a developing tropical country (*Hongsuwan et al., 2014*; *Nhu et al., 2014*). The high mortality observed in MDR *Acinetobacter* spp. bacteraemia is because treatment options are limited and those available are associated with toxicity (*Fishbain and Peleg, 2010*). The high proportions of *S. aureus*, *E. coli* and *K. pneumoniae* bactaeremia being caused by MRSA and *E. coli* and *K. pneumoniae* non-susceptible to extended-spectrum cephalosporins, respectively, are consistent with previous reports from other tropical countries (*Moreno et al., 2006*; *Cuellar et al., 2008*; *Rosenthal et al., 2003*). The rising proportions of community-acquired *E. coli* and *K. pneumoniae* bacteraemia being caused by *E. coli* and *K. pneumoniae* non-susceptible to extended-spectrum cephalosporins, and the rising proportion of hospital-acquired *Acinetobacter* bacteraemia being causing *Acinetobacter* non-susceptible to carbapenem suggest that the burden of AMR in Thailand is deteriorating over time.

A limitation of this study is that more complete clinical data were not available. Mortality attributable to MDR could be overestimated if MDR infection was associated with more severely ill patients in ICUs. However, our estimated attributable mortality is comparable to the previous reports. For example, our estimated mortality attributable to MDR *Acinetobacter* bacteraemia (40.6%) is comparable to the mortality attributable to imipenem resistant *Acinetobacter* bacteraemia reported by Kwon *et al.* in Korea (41.1%), which was adjusted by severity of illness (*Kwon et al., 2007*; *Falagas and Rafailidis, 2007*). In addition, data on hospitalization in other hospitals not participating in the study (for example, a smaller community hospital or a private hospital in the province) were not available, which could have resulted in a misclassification of CAB, HCAB and HAB in some cases. We also note that data on attributable mortality from different countries is difficult to compare because of the differing study designs. For example, our mortality outcome is the overall 30-day mortality, including both directly and indirectly contributed to MDR, while an EU study only considered directly attributable deaths (*European Centre for Disease Prevention and Control and European Medicines Agency, 2009*). The p values for trends were generated by the stratification method; therefore, the analysis was not biased towards the increasing availability of the hospital data over the study period. Nonetheless, the trends could be affected by an increasing use of blood culture, changes in antimicrobial agents tested for susceptibility, and greater standardization of laboratory methodologies over time (*Opartkiattikul and Bejrachandra, 2002*). It is likely that the burdens of MDR similar to that observed in our study are present in many secondary and tertiary hospitals in tropical LMICs, particularly where extended-spectrum cephalosporins and carbapenem are widely used. Nonetheless, resources for diagnostics, methodologies used in the laboratories, and study designs need to be carefully considered when performing a comparison between different settings.

Despite the increasing global focus on AMR in LMICs, considerable gaps remain in our understanding of the scale of the problem. We have demonstrated that the integration of information from readily available routinely collected databases can provide valuable information on the trends and mortality attributable to AMR in Thailand. The methodology used in our study could be applied to explore the burden of AMR in other LMICs where microbiological facilities and hospital admission database are available.

## Materials and methods

### Study population

From 2004 to 2010, Thailand was classified as a lower-middle income country with an average income of $4782 per person per year in 2010 (*WorldBank, 2015*). Northeast Thailand consists of 20 provinces covering 170,226 km and had an estimated population of 21.4 million in 2010. A large proportion of the population in this area lives in rural settings, with most adults engaging in agriculture with an emphasis on rice farming. Healthcare in Thailand is mainly provided by government-owned hospitals. Each province has a provincial hospital, which provides services and care to individuals within its catchment area. Additionally, provincial hospitals act as referral hospitals for smaller community hospitals for severely ill patients. All provincial hospitals receive comparable resources, which are proportional to the respective populations of the provinces. Provincial hospitals, unlike smaller community hospitals, are equipped with a microbiology laboratory capable of performing bacterial culture using standard methodologies for bacterial identification and susceptibility testing provided by the Bureau of Laboratory Quality and Standards, Ministry of Public Health (MoPH), Thailand (*Opartkiattikul and Bejrachandra, 2002*). During the study period, antimicrobial susceptibility was determined in all study hospitals using the disc diffusion method according to Clinical and Laboratory Standards Institute (CLSI) (*National Committee for Clinical Laboratory Standards, 2004*).

### Study design

We conducted a retrospective, multicentre surveillance study of all provincial hospitals in Northeast Thailand. From the hospitals that agreed to participate, data were collected from microbiology and hospital databases between January 2004 and December 2010. Hospital number (HN) and admission number (AN) were used as a record linkage between the two databases and to identify individuals who had repeat admissions. The death registry for Northeast Thailand was obtained from the Ministry of Interior (MoI), Thailand, and used to identify patients who were discharged from hospital but died at home shortly after, which is a common practice in Thailand (*Kanoksil et al., 2013*; *Hongsuwan et al., 2014*). Ethical permission for this study was obtained from the Ethical and Scientific Review Committees of the Faculty of Tropical Medicine, Mahidol University, and of the MoPH, Thailand. Written consent was given by the directors of the hospitals to use their routine hospital database for research. Consent was not sought from the patients as this was a retrospective study, and the Ethical and Scientific Review Committees approved the process.

### Data collection

The microbiology laboratory data collected included hospital number (HN), admission number (AN), specimen type, specimen date, culture result, and antibiotic susceptibility profile (antibiogram). We consulted with study sites when the results of antimicrobial susceptibility testing were unclear. Hospital admission data were collected from the routine in-patient discharge report, which is regularly completed by attending physicians and reported to the MoPH, Thailand, as part of the national morbidity and mortality reporting system. The data collected included HN, AN, national identification 13-digit number, admission date, and discharge date. Date of death was also extracted from this record. Data collected from the national death registry obtained from the MoI included the national identification 13-digit number and the date of death.

### Definitions

Bacteraemia was classified as CAB, HAB and HCAB as described previously (*Kanoksil et al., 2013*; *Hongsuwan et al., 2014*). Polymicrobial infection was defined in patients who had more than one species of pathogenic organisms isolated from the blood during the same episode, and was excluded from the analysis. Information on the incidence of CAB, HCAB and HAB from all pathogenic organisms has been published previously (*Kanoksil et al., 2013*; *Hongsuwan et al., 2014*).

The 30-day mortality of CAB and HCAB was defined as death within 30 days of the admission date. The 30-day mortality of HAB was determined on the basis of a record of death within 30 days of the positive blood culture taken as recorded in the routine hospital database or by a record of death in the national death registry. In the event that a patient had more than one episode of bacteraemia, only the first episode was included in the study.

The standard definition of MDR proposed by ECDC/CDC was used (*Magiorakos et al., 2012*). In brief, MDR was defined as non-susceptibility to at least one agent in three or more antimicrobial categories. Additionally, methicillin-resistant *Staphylococcus aureus* (MRSA) were automatically described as MDR (*Magiorakos et al., 2012*).

## Statistical analysis

Pearson's chi-squared test and Fisher's Exact test were used to compare categorical variables. A non-parametric test for trends was used to assess changes in prevalence of antimicrobial resistance over time stratified by hospital (using the npt_s command in STATA).

Mortality of patients with a first episode of HAB, HCAB and HAB caused by *S. aureus, Enterococcus* spp., *E. coli, K. pneumoniae, Pseudomonas aeruginosa*, and *Acinetobacter* spp. were evaluated in relation to MDR. We selected these organisms based on guidelines for MDR proposed by ECDC/CDC, (*Magiorakos et al., 2012*) and the fact that *E. coli* and *K. pneumoniae* were the most common causes of bacteraemia caused by *Enterobacteriaceae* in our setting (*Kanoksil et al., 2013*; *Hongsuwan et al., 2014*). Isolates tested for less than three antimicrobial categories were excluded from the analysis because they were not applicable to ECDC/CDC standard definitions of MDR. To examine the association between MDR and mortality, we performed a multivariable logistic regression analysis adjusting for a priori selected baseline confounders. To take account of the fact that patients with CAB, HCAB, and HAB were different populations with different definitions of 30-day mortality, we applied models to each group (CAB, HCAB and HAB) separately. Multivariable logistic regression models were developed using a purposeful selection method (*Bursac et al., 2008*). Potential confounding variables evaluated included age, gender and admission year. In the model for HAB, time to bacteraemia was also evaluated as a potential confounder because there was evidence suggesting that time to HAI was associated with mortality from HAI (*Moine et al., 2002*; *Nguile-Makao et al., 2010*). Time to bacteremia was defined as the duration between hospital admission and the date positive blood culture was taken. All models were stratified by hospital.

The mortality attributable to MDR was calculated using adjusted odds ratios (aORs) estimated by the final multivariable logistic regression models. If X was the observed mortality in patients with MDR infection, the estimated odds of mortality if they were infected with non-MDR organisms (O) would be (1/aOR)*(X/(1-X)). Assuming that excess mortality was due to MDR, then the mortality attributable to MDR would be the absolute difference between mortality in patients with MDR infection (X) and the predicted mortality if they were infected with non-MDR organisms (O/(1+O)), which would be X – (O/(1+O)) (*Benichou, 2001*; *Greenland and Robins, 1988*). Heterogeneity between different organisms within each group of patients (CAB, HCAB, and HAB) was assessed using the chi-squared test, and the percentage of variation due to heterogeneity (I-square) was calculated.

The number of deaths attributable to MDR in Thailand was determined using the methodology described previously (*European Centre for Disease Prevention and Control and European Medicines Agency, 2009*). Data used included our estimated mortality attributable to MDR bacteraemia and cumulative incidence of HAI bacteraemia, lower respiratory track infection (LRTI), urinary tract infection (UTI), skin and soft tissue infection (SSTI), and other sites of infection caused by MDR *S. aureus, E. coli, K. pneumoniae, P. aeruginosa*, and *Acinetobacter* spp. in Thailand in 2010, which have been described previously (*Pumart et al., 2012*). Death attributable to MDR *Enterococcus* spp. was not included as the cumulative incidence of MDR *Enterococcus* infection in Thailand was not available (*Pumart et al., 2012*). Attributable mortality by site of infection (LRTI, UTI, SSTI and other site) was estimated by applying correction factors corresponding to the relative mortality from infections of those sites compared to bacteraemia (*Martone et al., 1998*). All analyses were performed using STATA version 14.0 (StataCorp LP, College station, Texas, USA).

## Acknowledgements

We gratefully acknowledge the Bacterial Infection in North East Thailand (BINET) network for providing microbiological and hospital admission data. BINET are comprised of Chalavit Limpavithayakula, Chatchawan Namsorn, Chadaporn Paungmala (Amnat Chareon hospital), Chan Tantiwaraporn, Somsak Kitisriworapun, Kanyapak Panjaiya, Kridsada Sirichaisit (Bueng Kan hospital), Chalit Thongprayoon, Somchai Jittai, Rattana Jirajaruporn (Burirum hospital), Sompong Charoenwat, Some Prasarn, Wiset Phosawang, Peeraphorn Lamworaphong (Chaiyaphum hospital), Pramoth Boonjian,

Pattarawadee Wongmeema, Sunthorn Thongbai (Loei hospital), Sunthorn Yontrakul, Piyalak Siritaratiwat, Supanne Pisuttimarn (Mahasarakham hospital), Udom Pedphuwadee (Mukdahan hospital), Somkid Suriyalert, Pichai Thongtaradol, Somsri Viriyaphun (Nakhon Phanom hospital), Kittisak Danwiboon, Kanjana Kehatan, Wannakorn Dechnorasing (Nongkhai hospital), Narong Uen- trakul (Roi Et hospital), Somsak Chaosirikul, Anek Jaddee, Soontorn Romaneeyapech (Sisaket hospital), Thongchai Triwiboonwanich, Saowarat Deekae, Vorathep Wuttisil (Surin hospital), Pramot Srisamang, Nittaya Teerawattanasook (Ubon Ratchathani hospital), Pichart Dolchalermyuttana, Yaowalau Lootrakul, Sayrung Sripavatakul (Udon Thani hospital), Charan Thongthap, Pongsathorn Siripoulsak, Chainakhon Kaewluang (Yasothon hospital). We thank the Ministry of Interior, Thailand, for national death registry data. We thank Mayura Malasit and Jittana Suwannapruk for their administrative support. We gratefully acknowledge the support provided by staff at the Mahidol-Oxford Tropical Medicine Research Unit and at all participating hospitals. We thank Prapass Wanapinij, Kraiwuth Sriporamanont, Mongkol Fungprasertkul and Dean Sherwood for computational support. The authors thank NJ White for comments on the final draft.

## Additional information

### Funding

| Funder | Grant reference number | Author |
|--------|------------------------|--------|
| Ministry of Public Health | | Direk Limmathurotsakul |
| Wellcome Trust | 100484/Z/12/Z | Cherry Lim<br>Nicholas PJ Day<br>Direk Limmathurotsakul |
| Wellcome Trust | 101103/Z/13/Z | Direk Limmathurotsakul |

The funders had no role in study design, data collection and interpretation, or the decision to submit the work for publication.

### Author contributions

CL, ET, VW, SJP, Conception and design, Analysis and interpretation of data, Drafting or revising the article; MH, Acquisition of data, Analysis and interpretation of data; VT, SH, Analysis and interpretation of data, Drafting or revising the article; NPJD, Conception and design, Drafting or revising the article; DL, Conception and design, Acquisition of data, Analysis and interpretation of data, Drafting or revising the article

### Author ORCIDs

Cherry Lim, http://orcid.org/0000-0003-2555-6980
Nicholas PJ Day, http://orcid.org/0000-0003-2309-1171
Sharon J Peacock, http://orcid.org/0000-0002-1718-2782
Direk Limmathurotsakul, http://orcid.org/0000-0001-7240-5320

## Additional files

### Supplementary files

• Supplementary file 1. Factors associated with 30-day mortality of bacteraemia patients.

• Supplementary file 2. Parameters used to estimate mortality attributable to multidrug-resistance in Thailand.

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
