## [Decision Letter]

Thank you for submitting your article "Epidemiology and Burden of Multidrug-resistant Bacterial Infection in a Developing Country" for consideration by *eLife*. Your article has been reviewed by three peer reviewers, one of whom is a member of our Board of Reviewing Editors and the evaluation has been overseen by Prabhat Jha as the Senior Editor. The reviewers have opted to remain anonymous.

The reviewers have discussed the reviews with one another and the Reviewing Editor has drafted this decision to help you prepare a revised submission.

Summary:

This manuscript makes an important contribution on mortality from multidrug-resistant (MDR) bacterial infection in low- and middle-income countries (LMIC). Limited mortality data from MDR bacterial infection exists in these settings. The lack of antimicrobial stewardship and poor, if any, over-the-counter control is a great concern in the face of the global health threat of multidrug resistance. Although retrospective, this is a well designed study including not only laboratory data but also some clinical data (admission and death registry) covering a large number of hospitals over 7 years.

Essential revisions:

1) The data covers the period from 2004 onwards, with different hospitals starting records at different times. It seems that Table 1–Table 4 have simply aggregated these results over time, but the authors should comment on how trends in data availability might affect the results. E.g. will increasing availability favour statistical significance amongst the results that were collected in later years? Were there differences over time either because of enhanced data collection or diagnostics? If there were no apparent trends in percent-MDR over this period, it would be helpful to show as supporting information (to illustrate that the overall epidemiology might be reasonably stable).

2) How generalizable are the sampled hospitals? Are they comparable in size, populations, and resources? Are there inter-hospital movement of participants? How common is this and how does this impact findings?

3) Expressing the estimated number of deaths in HAI attributable to MDR bacteria in percentage (in brackets after the absolute numbers) would read better.

4) There needs to be more clarification around the precise denominators being used, for the results. In particular:

a) The wording 'percentage of MDR', used throughout the manuscript, is pretty confusing – it suggests the proportion of all MDR cases (including in the community) that cause CAB, etc. This can't be possible – Instead it seems that the authors are referring to the inverse: the proportion of CAB being caused by MDR.

b) However, even this needs more clarification: as the denominators in Table 1 are not the same down each column, it seems what the authors are referring to is the proportion of (e.g.) CAB caused by a particular pathogen, that are caused by MDR variants of that pathogen.

c) If this correct, the language should be adjusted throughout the manuscript, to make this denominator very clear – suggest something like: "Of CAB, HCAB and HAB caused by *S. aureus*, the percentage being caused by MDR *S. aureus* were…", and removing references to "The proportion of MDR" throughout.

5) When comparisons are drawn with other settings take into account differences in setting, resources, study design, and other characteristics that may introduce bias and preclude a direct comparison.

---

## [Author Response]

Essential revisions:

1) The data covers the period from 2004 onwards, with different hospitals starting records at different times. It seems that Table 1–Table 4 have simply aggregated these results over time, but the authors should comment on how trends in data availability might affect the results. E.g. will increasing availability favour statistical significance amongst the results that were collected in later years? Were there differences over time either because of enhanced data collection or diagnostics?

The following sentence has been added in the Discussion section for clarity, “The p values for trends were generated by the stratification method; therefore, the analysis was not biased towards the increasing availability of the hospital data over the study period. Nonetheless, the trends could be affected by an increasing use of blood culture, changes in antimicrobial agents tested for susceptibility, and greater standardization of laboratory methodologies over time.”

If there were no apparent trends in percent-MDR over this period, it would be helpful to show as supporting information (to illustrate that the overall epidemiology might be reasonably stable).

Figures were added as suggested.

2) How generalizable are the sampled hospitals? Are they comparable in size, populations, and resources? Are there inter-hospital movement of participants? How common is this and how does this impact findings?

The following sentence has been added in the Discussion section for clarity, “It is likely that burdens of MDR similar to that observed in our study are present in many secondary and tertiary hospitals in tropical LMICs, particularly where extended-spectrum cephalosporins and carbapenem are widely used. Nonetheless, resources for diagnostics, methodologies used in the laboratories, and study designs need to be carefully considered when performing a comparison between different settings.”

3) Expressing the estimated number of deaths in HAI attributable to MDR bacteria in percentage (in brackets after the absolute numbers) would read better.

The percentage in brackets, “(43%)”, has been added as suggested.

*4) There needs to be more clarification around the precise denominators being used, for the results. In particular:*

*a) The wording 'percentage of MDR', used throughout the manuscript, is pretty confusing – it suggests the proportion of all MDR cases (including in the community) that cause CAB, etc. This can't be possible – Instead it seems that the authors are referring to the inverse: the proportion of CAB being caused by MDR.*

*b) However, even this needs more clarification: as the denominators in Table 1 are not the same down each column, it seems what the authors are referring to is the proportion of (e.g.) CAB caused by a particular pathogen, that are caused by MDR variants of that pathogen.*

c) If this correct, the language should be adjusted throughout the manuscript, to make this denominator very clear – suggest something like: "Of CAB, HCAB and HAB caused by S. aureus, the percentage being caused by MDR S. aureus were…", and removing references to "The proportion of MDR" throughout.

The reviewers are correct; it is actually the proportion of CAB being caused by MDR variants of that pathogen. We have carefully edited this in the manuscript thoroughly as suggested.

5) When comparisons are drawn with other settings take into account differences in setting, resources, study design, and other characteristics that may introduce bias and preclude a direct comparison.

The following sentence has been added in the Discussion section for clarity, “Nonetheless, resources for diagnostics, methodologies used in the laboratories, and study designs need to be carefully considered when performing a comparison between different settings.”